# Gestational Diabetes Mellitus and Its Correlation in the Development of Pancreatic Cancer: A 10-Year Systematic Review

**DOI:** 10.3390/cancers16101840

**Published:** 2024-05-11

**Authors:** Sophia Tsokkou, Ioannis Konstantinidis, Maria-Nefeli Georgaki, Dimitrios Kavvadas, Kyriaki Papadopoulou, Antonios Keramas, Antonia Sioga, Theodora Papamitsou, Sofia Karachrysafi

**Affiliations:** 1Research Team “Histologistas”, Interinstitutional Postgraduate Program “Health and Environmental Factors”, Department of Medicine, Faculty of Health Sciences, Aristotle University of Thessaloniki, 54124 Thessaloniki, Greece; stsokkou@auth.gr (S.T.); ikonsc@auth.gr (I.K.); mgeorgaki@cheng.auth.gr (M.-N.G.); kavvadas@auth.gr (D.K.); kyriakinp@auth.gr (K.P.); antonios@auth.gr (A.K.); sioga@auth.gr (A.S.); thpapami@auth.gr (T.P.); 2Laboratory of Histology-Embryology, Department of Medicine, Faculty of Health Sciences, Aristotle University of Thessaloniki, 54124 Thessaloniki, Greece; 3Environmental Engineering Laboratory, Department of Chemical Engineering, Aristotle University of Thessaloniki, 54124 Thessaloniki, Greece

**Keywords:** gestational diabetes mellitus, pancreatic cancer, adenocarcinoma

## Abstract

**Simple Summary:**

The purpose of this systematic review was to investigate the correlation between gestational diabetes mellitus—the diabetes women experience during pregnancy—and the development of pancreatic cancer. Thus, for this research a 10-year literature review was conducted using the keywords “gestational diabetes mellitus”, “pancreatic cancer” and “adenocarcinoma” in research databases such as PubMed, Scopus and ScienceDirect. After the exclusion criteria were added and the duplicates were removed, the articles were examined for their eligibility, and those suited for the study based on the degree of relevance were included. From the articles selected, the necessary data were extracted into tables, and the necessary information was summarized. Furthermore, a quality assessment was performed based on the type of study found, which revealed a high level of eligibility and related content, making the articles sufficient for us to conclude that gestational diabetes mellitus and pancreatic cancer are correlated.

**Abstract:**

Purpose: Pancreatic cancer (PC) is a fatal malignancy with an aggressive course derived from the cells of pancreatic tissue. Gestational diabetes mellitus (GDM) is a state of spontaneous hyperglycemia occurring during gestation and has been suggested as a risk factor PC. Women with a history of GDM revealed a risk rate of 7.1% for the development of PC. The current systematic review aims to investigate the correlation between GDM and the degree to the prevalence of PC. Methodology: For this systematic review, the PICO model was prepared to construct and outline the exact questions of the study, a PRISMA flow diagram was prepared and quality assessment was conducted using the Newcastle Ottawa Scale (NOS) for Cohort Studies, the NIH Quality Assessment Tool-Criteria for Case Reports and the Cochrane quality assessment tool for Systematic Reviews and Meta-analysis studies. Result: A total of eight articles were retrieved from the main databases, and a table was created to summarize the information found. Even though the data found were limited, the quality assessment performed revealed that the articles were of high validity. Conclusions: It can be concluded that GDM has an association with the development of PC and can be considered as a risk factor.

## 1. Introduction

Recent studies have shown high rates of development of pancreatic cancer (PC), especially in European countries, where its rise is predicted to be the second highest cause of mortality by 2030 [1]. Gestational diabetes mellitus (GDM) has been suggested as a risk factor for the development of PC. Women with a history of GDM revealed a risk rate of 7.1% for PC [2]. The current systematic review aims to investigate the correlation between GDM and the degree of the prevalence of PC.

### 1.1. Definition and Epidemiology of Pancreatic Cancer

Pancreatic cancer (PC) is a fatal malignancy with an aggressive course derived from cells of pancreatic tissue, mainly of the exocrine cells. PC is the tenth most common type of cancer and the third most common cause of cancer-related death in the United States, with an estimated 64,050 new diagnoses in the United States each year, accounting for 50,550 deaths in 2023 [3]. PC’s mortality rate in Europe reached 17 per 100,000 women in 2022 and 22.3 per 100,000 in men, revealing a greater prevalence in the male population [4]. PC is more common with increasing age and has a median age at death at 72 years old based on 2016–2020 cases [3,5]. Approximately 1.7% of men and women are diagnosed with PC at some point during their lifetime, based on 2017–2019 data [3,5]. The most common type of pancreatic cancer is pancreatic duct adenocarcinoma (PDAC), which is responsible for 90% of cases worldwide [6]. PC’s high mortality rate is attributed both to late diagnosis, since most patients do not develop symptoms until the disease reaches a more advanced stage, and to limited available responses to existing treatments [7].

#### Risk Factors of Pancreatic Cancer

Risk factors that are proved to be correlated with PC include both modifiable factors, such as heavy smoking, alcohol consumption, obesity, gut microbiome and Helicobacter pylori infection, and non-modifiable, such as age over 55 years, male sex, blood group, family history of diabetes, chronic pancreatitis cirrhosis of the liver, family history of PC and genetic susceptibility, which accounts for 5–10% of newly-diagnosed cases [8,9]. Diabetes mellitus of recent onset has been found to be one of the clinical manifestations of PC in 68% of the patients in a recent study [10] (Figure 1).

### 1.2. Diabetes Mellitus

#### 1.2.1. Diabetes Mellitus and Its Classification

By definition, diabetes mellitus (DM) is characterized as a “metabolic disease with highly elevated glucose levels in the bloodstream” [12]. In other words, it is a heterogeneous group of metabolic disturbances in the processes of the glucose metabolism with the key finding being chronic elevated blood glucose concentration (hyperglycemia) as a result of impaired secretion of insulin from pancreatic β-cells, impaired mechanisms of insulin action or both [13]. There are many classifications of DM including type 1 diabetes mellitus (T1DM), type 2 diabetes mellitus (T2DM), gestational diabetes mellitus (GDM), maturity-onset diabetes of the young (MODY), neonatal diabetes mellitus, Wolfram syndrome, type 3c diabetes, steroids-induced diabetes, cystic fibrosis diabetes, Alström Syndrome and latent autoimmune diabetes in adults (LADA) [14]. The three most common types are T1DM, T2DM and GDM. Firstly, T1DM is classified as an autoimmune disease caused by the destruction of β-cells that are responsible for the production of insulin, resulting in the long-term administration of insulin through daily injections, insulin pump therapy and automated insulin delivery systems [15]. Secondly, T2DM is considered the most common type of DM, and it accounts for 90% of cases. It is based on insulin resistance, especially in individuals of age greater than 45. However, in recent years, T2DM has become an increasing issue in children, adolescents and young adults. Thirdly, GDM is the state of spontaneous hyperglycemia occurring during gestation, commonly during the second and third trimesters of pregnancy [16]. It is referred to as a transient state of impaired glucose tolerance solely affecting women throughout pregnancy. Based on the American Diabetes Association (ADA), 10% of pregnancies are affected by GDM annually [17], but according to the most recent (2021) International Diabetes Federation (IDF) Diabetes Atlas, GDM affects approximately 21.1 million live births or 16.7% pregnancies worldwide annually [18].

#### 1.2.2. Pathophysiology of Gestational Diabetes Mellitus

The cause of GDM is the result of various components including genetic and environmental factors, as well as hormonal contributors. During the gestational period, the increased concentration of progestins and estrogens—pregnancy-related hormones—results in slower gastric emptying and lower fasting glucose concentration. Thus, the tissues start to become less sensitive to insulin, leading to an increase in the postprandial glucose concentration (ideal values for women with GDM are fasting glucose ≤5–5.3 mmol/L (90–95 mg/dL) and either one-hour post-meal ≤ 7.8 mmol/L (140 mg/dL) or 2 h post-meal ≤ 6.7 mmol/L (120 mg/dL)) [19,20]. Under normal circumstances, pregnant women should secrete adequate insulin through pancreatic β-cells to make up for the difference in sensitivity of tissues, resulting in an equilibrium of a normoglycemic state; however, women with GDM are not able to secrete adequate insulin to compensate for the resistance, and thus insulin B receptors are unable to undergo tyrosine phosphitylation [21] (Figure 2).

#### 1.2.3. Gestational Diabetes Mellitus Diagnosis

Testing for GDM is a standard screening procedure during the gestational period aiming for the early detection and regulation of affected women [22]. The American Diabetes Association (ADA) suggests that women should be screened between the 24th and 28th weeks of gestation unless they are in a low-risk category [17,22]—women age 25 and younger, of a low-risk race, with normal weight gain and no history of macrosomia [23]. There are two recommended approaches for screening during pregnancy during the 24th to 28th weeks. In the first approach, women are screened in a two-step process by measuring the plasma glucose levels 1 h after administration of 50 g of glucose. Women with a glucose concentration greater or equal to 130 and 140 undergo a 100 gm glucose tolerance test (OGTT) on another day. In the second approach, women are directly tested with 100 gm OGTT. The diagnosis is established by the Carpenter and Coustan criteria (Table 1). Firstly, venous blood is drawn when women are still in the fasting period; after glucose is provided to the women, venous blood is drawn every 1 h consecutively until 3 h are reached [23,24].

#### 1.2.4. Gestational Diabetes Mellitus—Maternal and Fetal Complications

GDM can have potential short- and long-term complications for both the mother and the infant. GDM can result in preterm births, leading, mainly, to respiratory distress syndrome due to insufficient or absent surfactant from the improper type II pneumonocytes and large for gestational age infants—fetal weight greater than 4000 gr, which increases the risk of proceeding with a C-section to avoid complications caused during vaginal birth, such as shoulder dystocia [25,26]. Other fetal complications include hyperbilirubinemia, neonatal hypoglycemia and hypocalcemia, polycythemia and increased perinatal mortality [27]. Regarding the mother, GDM increases the risk for high blood pressure, which, as a result, increases the likelihood for pre-eclampsia—a life-threatening complication for both the mother and the infant [26]. Studies suggest that women with a history of GDM have a greater risk in developing T2DM compared to women with normoglycemic pregnancies [19]. The risks for the transition from GDM to T2DM ranges between 3% and 70%; however the likelihood rises with women giving birth after the age of 30, birth weight of babies delivered being greater than 3.5 kg and undergoing insulin treatments throughout the gestational period [19,22]. Both T1DM and T2DM have been found to be highly associated with the development of various cancer types, with PC being one of them; however not a lot of studies have addressed the extent of the correlation between GDM and PC. In the current systematic review, the correlation between GDM and PC is examined, Figure 3.

## 2. Materials and Methods

### 2.1. PICO Model

For this systematic review, the PICO model was used to construct and outline the exact question of the study (Table 2), and the Preferred Reporting Items for Systematic Reviews and Meta-Analysis (PRISMA) guidelines were used for the extraction, screening and assessment of the relevant articles found. Additionally, quality assessment tables were prepared depending on the type of studies found, including, the Newcastle Ottawa Scale (NOS) for Cohort Studies, the NIH Quality Assessment Tool-Criteria for Case Reports and the Cochrane quality assessment tool for Systematic Reviews and Meta-analysis studies. The study has not been registered in PROSPERO.

### 2.2. PRISMA Flow Diagram

A PRISMA flow diagram was prepared using the code based on the keywords “((“pancreatic” OR “pancreas”) AND (“adenocarcinoma” OR “carcinoma” OR “adenocarcinomas” OR “carcinomas” OR “cancer” OR “cancers”)) AND ((“gestational diabetes mellitus”) OR (“GDM”))”. The inclusion criteria included articles published within a 10-year duration (2014–2024), and review articles of the categories in the literature, both systematic reviews and meta-analyses, as well as case reports were chosen. The articles had to be written in English and be available in the form of open access. The databases used were PubMed, Scopus and ScienceDirect. Initially, the total number of all articles found without the inclusion criteria were recorded. After the criteria were applied, the automation tools excluded the irrelevant articles, and the remaining articles were screened. Duplicated articles were excluded, and the articles were assessed and retrieved based on the degree of relevance from their titles, abstract and main-text information (Figure 4). A table was then prepared, including all relevant articles that were assessed and accepted as appropriate for this systematic review (Table 3).

### 2.3. Methodological Quality Assessment

Moreover, a quality assessment was performed to examine and evaluate the accuracy and quality of each article. The Newcastle Ottawa Scale (NOS) was used to evaluate the quality of cohort studies—with the lowest score being 0 and the highest being 8. The NIH quality assessment tool was used for the evaluation of case reports—0 being the lowest and the highest being 9. Lastly, the Cochrane quality assessment tool was used for systematic reviews and meta-analysis articles—with scores of 0 being the lowest and 7 the highest (Table 4, Table 5 and Table 6).

## 3. Results

### 3.1. PRISMA Flow Diagram

Firstly, the total number of articles found without applying the criteria was recorded for all three databases. The total number of articles found was 1128, with 17 being from PubMed, 18 from Scopus and 1093 from ScienceDirect. After the criteria were applied—10-year publication window, review articles (literature, systematic and meta-analyses) and case reports, open access and published in the English language—997 articles were excluded as ineligible by the automation tools. Thus, 131 articles were recorded and screened. Moving on, 3 articles were excluded as they were duplicates, and 128 were evaluated. From those, a total of 121 articles were excluded for the following reasons—83 articles were excluded due to low relevance based on their titles, 30 from the content of their abstracts and 8 from the content of their main text. The articles were evaluated using the process of double-blind assessment—being evaluated by two different reviewers independently to eliminate bias of decision. One article was included from citations and a total of eight articles were included.

A table [Table 2] was then prepared to summarize the finding of those eight articles with the following categories: authors, risk factors of development of GDM, risk factors of development of PC, identification of GDM and, lastly, association between GDM and PC.

### 3.2. Methodological Quality Assessment

Additionally, three different quality assessments were prepared including the Newcastle Ottawa Scale (NOS), the NIH quality assessment criteria tool and the Cochrane risk bias tool.

#### 3.2.1. Newcastle Ottawa Scale (NOS)

Firstly, the NOS is used to assess the quality and validity of non-randomized studies (case control and cohort studies). For this systematic review, two cohort studies—Peng et al. [31] and Simon J. [7]—were examined using the scale of 0 to 7 (0 being the lowest quality and 7 the highest). For the Selection category, both studies clearly stated the goal of their study and included an adequate sample size to support the research they wanted to conduct. Regarding the comparability, both studies included a control group for accurate comparison with the study group. Moreover, both groups included possible risk factors that might interfere or be a complication of GDM. However, for the Peng et al. [31] study, no statistical test was provided in comparison with the Simon J. [7] study, and Simon J.’s [7] study did not provide sufficient information on the outcome regarding the correlation between GDM and PC. Both studies received a score of 7 out of 8, suggesting that the quality and validity of their study accurate (Table 4).

#### 3.2.2. NIH Quality Assessment Tool

The NIH quality assessment tool was used for the evaluation and methodological assessment of case reports. The score range is from 0 to 9; a score of 0 to 3 is classified as “poor”, 4 to 6 as “fair” and 7 to 9 as “good”. In this systematic review, two case reports were examined, including the studies of Quaresma P. and Shi A.W. [30,33]. In both studies, the research question was clearly stated, and the population sample was clearly defined. In the study of Quaresma P. [33], the participation rate was considered eligible as more than 50% of the population remained until the end of the study, and there was a uniform eligibility across all participants. The study received a total score of 4, entering the “fair” category. The Shi A.W. [30] study included a uniform eligibility criterion for all participants and evaluated the different levels of exposure among the population. The independent variables clearly defined and implemented consistently and thus the study received a total score of 5, again falling into the “fair” category. The studies overall received a moderate score (Table 5).

#### 3.2.3. Cochrane Risk Bias Tool

For the Cochrane risk bias tool, three systematic reviews and meta-analysis studies were included: Tong G.-X., Wang Y. and Slouha E [29,32,35]. Six domains were used to classify and rate the context of each study with the following information, random sequence generation, allocation concealment, participant and personnel blinding, outcome assessment blinding, incomplete outcome data and biased reporting and further biased sources. The bias was assessed using the following criteria: high, low and unclear. The study of Tong G.-X [29] received the highest score of the three, which was 6, as it included all relevant information. The study of Slouha E. [35] received a score of 5, and the study of Wang Y. [32] received the lowest score of 2 (Table 6).

## 4. Discussion

Based on the systematic research conducted, data collected using the PRISMA guidelines and PICO model, the following results can be stated from the information summarized in Table 3.

### 4.1. Risk Factors for the Development of Gestational Diabetes Mellitus

Firstly, three out of the eight studies, namely Slouha E, Peng et al. and Wang Y, [31,32,35], concluded that increased maternal age increases the risk of development of GDM, and four out of eight studies, namely Slouha E, Peng et al., Quaresima, Wang Y and Tong G.-X., [29,31,33,35], stated that higher maternal weight and obesity are contributors to higher risks in the development of GDM. Based on the studies retrieved, it can be stated that the reason for this correlation might be due to higher nutrient intake, greater circulating adipokine and greater oxidative stress among pregnant women in advanced maternal stage [36]. The studies of Wang Y. and Slouha E. [32,35], stated that except for obesity and increased maternal age, PCOS as well as family history of T2DM, prediabetes and previous history of fetal death can also play an important role in the development of GDM [32].

### 4.2. Risk Factors for the Development of Pancreatic Cancer

Moreover, the study of Shi A.W. [30] states that the risk factors for PC are GDM, HELLP syndrome hemolysis, elevated liver enzymes and low platelets, a complication considered as a variant of pre-eclampsia [37], pulmonary hypertension and disseminated intravascular coagulation (DIC) in late pregnancy. Furthermore, the study of Quaresima P. [33] stated that obesity, heavy smoking, alcohol intake, history of diabetes, chronic pancreatitis, chronic cirrhosis, history of cholecystectomy and genetic predisposition for PC have an estimated rate of 5–10% of patients being diagnosed with PC. Additionally, HELLP syndrome was again identified as a risk factor. The study of Simon J. [7] states that the risk of PC increases by 80% in the case of type 2 diabetes.

### 4.3. Association of Gestational Diabetes Mellitus and Development of Pancreatic Cancer

When it comes to the association of GDM and the development of PC, the study of Tong, G.-X. [29] states that in the same manner that DM is related to PC, GDM can have the same impact. The study mentions five new pancreatic cancer cases being found in women with history of GDM and 49 without a GDM history, thus estimating the hazard ratio of GDM history being 7.1. The study of Shi A.W. [30] notes the same risk association showing that GDM can an important risk factor for the development of PC. Additionally, the study of Peng et al. [31] states that women with GDM had a higher risk of developing PC in comparison with women who did not suffer from GDM. However, the overall association was still low. Furthermore, the study of Wang Y. [32] showed that two cohort studies found a significant a positive association between GDM and risk of PC, whereas another cohort study did not. In the study by Quaresima P. [33], GDM showed a 7-fold increase in the risk of PC over the course of the women’s lives. Moreover, the study of Simon J. [7] concluded that over the eight years of follow-up, women with GDM were significantly associated with a higher risk of hospitalization with PC. Moving on, the study of Choudhury AA [34] reports that patients who have been diagnosed with DM are more likely to develop cancers such as pancreatic, colon, liver, kidney and bladder cancer, and a high linkage of GDM and development of cancer can be observed. Lastly, the study of Slouha E [35] states that women with GDM have higher chances of developing breast, ovarian, cervical and uterine cancer, as well as cancer in non-reproductive organs, like the thyroid and pancreas.

## 5. Conclusions

Thus, based on the data examined, it can be concluded that GDM has an association with the development of PC and can be considered as a risk factor. A major set-back to this study was the limited data found, as this topic is not widely examined. However, based on the data collected and the quality assessments performed, the studies found are credible and valid enough to support the correlation and linkage between GDM and the development of PC.

## Figures and Tables

**Figure 1 cancers-16-01840-f001:**
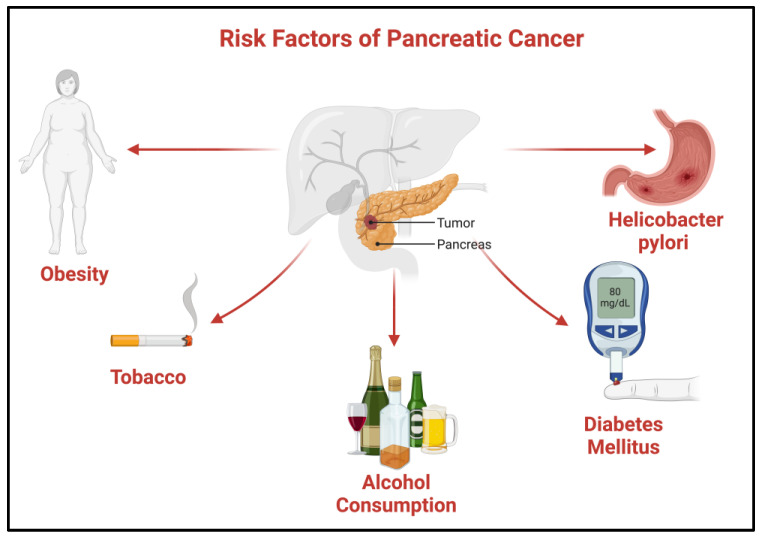
Risk factors of pancreatic cancer; created with BioRender.com [11].

**Figure 2 cancers-16-01840-f002:**
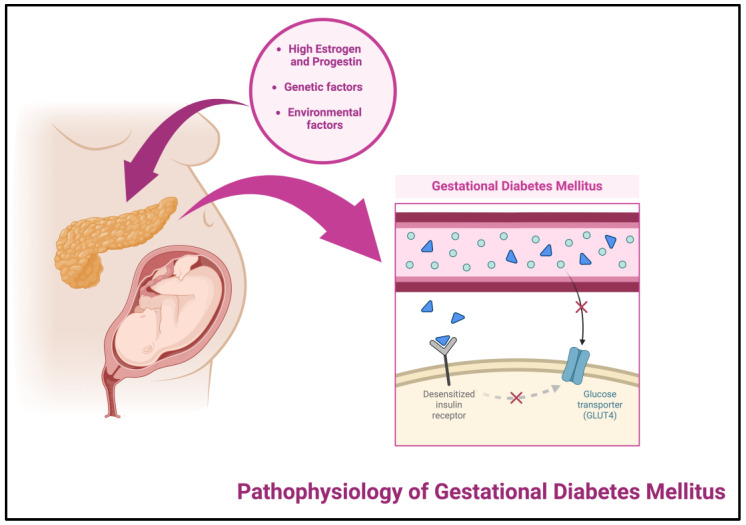
Pathophysiology of gestational diabetes mellitus; created with BioRender.com [11].

**Figure 3 cancers-16-01840-f003:**
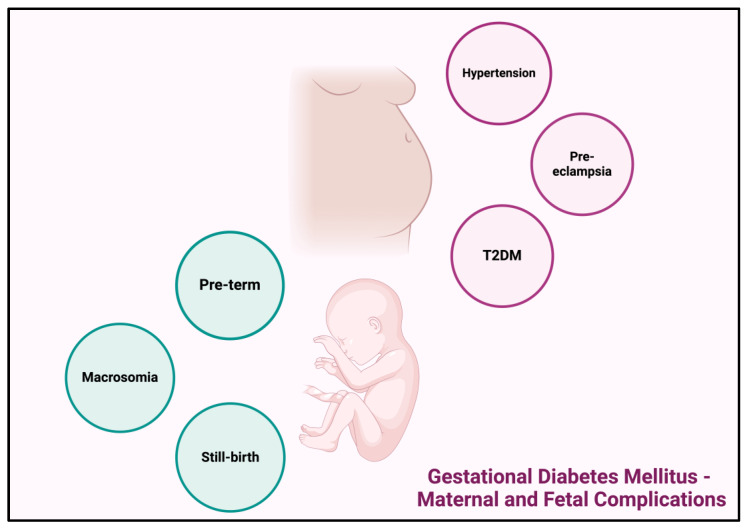
Maternal and fetal complications caused by gestational diabetes mellitus; created with BioRender.com [11].

**Figure 4 cancers-16-01840-f004:**
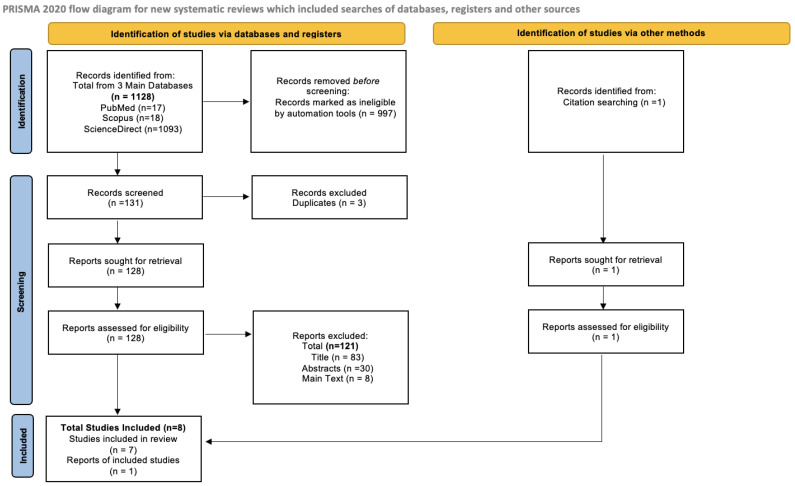
PRISMA Flow Diagram [28].

**Table 1 cancers-16-01840-t001:** Carpenter and Coustan criteria. If any 2 values are abnormal, then the patient is diagnosed as GDM.

Plasma Glucose Values (mg/dL)	
Fasting	95
1 h	180
2 h	150
3 h	140

**Table 2 cancers-16-01840-t002:** PICO table outlining Population (P), Intervention (I), Comparator (C) and Outcome (O).

PICO	
P	Pregnant women of no specific age or number of past pregnancies
I	Gestational diabetes mellitus
C	Pancreatic cancer
O	Gestational diabetes mellitus as a risk factor for the development of pancreatic cancer

**Table 3 cancers-16-01840-t003:** Table summarizing the articles retrieved based on the PRISMA flow diagram. Abbreviations: gestational diabetes mellitus (GDM), pancreatic cancer (PC), type 2 diabetes mellitus (T2DM), diabetes mellitus (DM), polycystic ovarian syndrome (PCOS).

#	Authors	Type of Study	Risk Factor for GDM	Risk Factor for PC	Identification of GDM	Association between PC and GDM
1	Tong, G.-X., 2014 [29]	Systematic Review	Obesity	-	The unexpectedly high prevalence of hyperglycemia among cancer patients led investigators to suggest the use of blood glucose measurements as a new screening or diagnostic method for cancer.	Similar to DM, GDM seems to be related to pancreatic cancer. Five new pancreatic cancer cases were found in women with GDM history and forty-nine without history; the hazard ratio of GDM history was 7.1. Similar results observed in two studies.
2	Shi AW, 2018 [30]	Case report	GDM is reported to be increasing and is more common among African Americans, Hispanics, Asians and Native Americans than among non-Hispanic whites	GDM, HELLP syndrome, pulmonary hypertension and DIC in late pregnancy.	Known GDM before admitted to hospital.	Women with a history of gestational diabetes showed a relative risk of pancreatic cancer of 7.1. Studies showed that gestational diabetes mellitus could be one of the important risk factors for pancreatic cancer.
3	Peng et al., 2019 [31]	Cohort Study	Phenomenon increases with maternal age, obesity issues and decreases with daily physical activity	-	-	Women with GDM had a higher risk of developing PC in comparison with women who did not suffer from GDM, but the overall association was still low.
4	Wang Y, 2020 [32]	Systematic review and Meta-analysis	PCOS, maternal obesity or overweight, family history of type 2 diabetes mellitus (T2DM), prediabetes, previous history of fetal death and increased maternal age	-	Mixed identification methods for GDM from studies retrieved. Either self-reported, OGTT or glucose challenge test.	Two cohort studies found a significant positive association between GDM and pancreatic cancer risk, whereas another cohort study did not. Evidence of severe heterogeneity.
5	Quaresima P, 2021 [33]	Case report and Literature review	-	Obesity, heavy smoking, alcohol intake, history of diabetes, chronic pancreatitis, chronic cirrhosis, previous cholecystectomy and genetic predisposition, with approximately 5–10% of patients diagnosed with a pancreatic cancer having a family history of the disease. Presented with HELLP syndrome.	-	GDM shows a 7-fold increase in the risk of pancreatic cancer over the course of their lives.
6	Simon J, 2021 [7]	Cohort study	-	Risk of PC increases by 80% in case of type 2 diabetes.	Before 2010, GDM screening was based on a two-step procedure for all women: the first test was on venous blood glucose 1 h after ingestion of 50 g of glucose, and in the event of a positive result, the second screening test was performed for oral glucose tolerance. Since 2010, recommended screening for these women is fasting blood glucose at the first prenatal consultation, and if it is not performed, an oral glucose tolerance test in the second trimester.	Over the eight years of follow-up, GDM was significantly associated with a higher risk of hospitalization with PC in the first and second Cox regression models adjusted for age and subsequent type 2 diabetes.
7	Choudhury AA, 2021 [34]	Literature Review	Insulin resistance, decreased chemerin levels, hereditary, reduced pancreatic insulin production similar to T2DM, obesity	-	Increased levels of glucose and C-reactive protein, lower levels of sex hormone-binding globulin and an increased chance of hyperinsulinemia when compared to pregnant women who do not have GDM.	It is reported that patients who have been diagnosed with DM are more likely to develop cancers such as pancreatic, colon, liver, kidney, bladder or BC. High linkage of GDM and development of cancer.
8	Slouha E, 2024 [35]	Systematic Review	Age, gestation, obesity and PCOS	-	Afamine, 1,5-anhydroglucitol and adiponectin markers.	Women with GDM have higher chances of developing breast, ovarian, cervical and uterine cancer and cancer in non-reproductive organs, like thyroid and pancreas.

**Table 4 cancers-16-01840-t004:** Assessment of methodological quality of cohort studies, according to the adapted Newcastle Ottawa Scale (NOS).

References	Selection				Comparability		Outcome		Total Quality Score
	Representativeness of Exposed Cohort	Sample Size	Ascertainment of Exposure	Non-Respondents	Adjust for the Most Important Risk Factors	Adjust for Other Risk Factors	Assessment of Outcome	Statistical Test	
Peng et al., 2019 [31]	1	1	1	1	1	1	1	0	7
Simon J, 2021 [7]	1	1	1	1	1	1	0	1	7

Methodological quality classification based on total score: <5: low quality; 5–7: moderate quality; >7 high quality.

**Table 5 cancers-16-01840-t005:** Assessment of methodological quality of case reports, according to the adapted NIH quality assessment tool.

Reference	NIH Quality Assessment Tool-Criteria									
	Q’1	Q’2	Q’3	Q’4	Q’5	Q’6	Q’7	Q’8	Q’9	Total Quality Score
Quaresima P, 2021 [33]	1	1	1	1	0	0	0	0	0	4
Shi AW, 2018 [30]	1	1	0	1	0	1	1	0	0	5

Question 1: Research question clearly stated. Question 2: Study population clearly defined. Question 3: Participation rate of eligible persons at least 50%. Question 4: Uniform eligibility criteria across all participants. Question 5: Sample size justification. Question 6: Evaluating different levels of exposure. Question 7: Independent variables clearly defined and implemented consistently. Question 8: Statistical methods were well-described and confounding variables considered. Question 9: Outcome measures were clearly defined and results well-described. Methodological quality classification based on total score: Good: 7–9 criteria; Fair: 4–6 criteria; Poor: 0–3 criteria.

**Table 6 cancers-16-01840-t006:** Reporting the quality/risk of bias with Cochrane risk bias tool.

Author and Year	Random Sequence Generation	Allocation Concealment	Participant and Personnel Blinding	Outcome Assessment Blinding	Incomplete Outcome Data and Biased Reporting	Further Biased Sources	Total Score
Tong, G.-X., 2014 [29]	+	+	+	+	+	+	6
Wang Y., 2020 [32]	+	-	?	?	+	-	2
Slouha E., 2024 [35]	+	+	?	+	+	+	5

Low risk of bias (+); High risk of bias (-); Unclear risk of bias (?).

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
