# Peer review of "Gestational Diabetes Mellitus and Its Correlation in the Development of Pancreatic Cancer: A 10-Year Systematic Review"

_cancers, 2024, doi:10.3390/cancers16101840_

Round 1

Reviewer 1 Report

Comments and Suggestions for Authors

There is no question about the link between diabetes mellitus, especially gestational diabetes mellitus, and the risk of pancreatic cancer. According to meta-analyses, those with newly diagnosed diabetes mellitus have a 5-7-fold increased risk of pancreatic cancer, especially in the first year following diagnosis. Between 50 and 74 percent of diabetes mellitus patients are linked to pancreatic cancer.
Perrin reports that during a follow-up period of 28–40 years from the time gestational diabetes mellitus was detected, the incidence of pancreatic cancer in women with the condition was 7.1 (95% confidence interval 2.8–18). (Perrin M, Terry M, Kleinhaus K. et al. Gestational diabetes as a risk factor for pancreatic cancer: a prospective cohort study. BMC Med 5, 25 (2007). https://doi.org/10.1186/1741-7015-5-25).

The review's statistics are provided in an easy-to-read manner, accurately capturing the dynamics of the rise in pancreatic cancer cases and their connection to diabetes mellitus.

Only one review paper on pancreatic cancer in relation to gestational diabetes has been published in the last five years, according to PubMed  (Quaresima P, Saccone G, Pellegrino R, Vaccarisi S, Taranto L, Mazzulla R, Bernardo S, Venturella R, Di Carlo C, Morelli M. Incidental diagnosis of a pancreatic adenocarcinoma in a woman affected by gestational diabetes mellitus: case report and literature review. Am J Obstet Gynecol MFM. 2021 Nov;3(6):100471. doi: 10.1016/j.ajogmf.2021.100471) and one piece of meta-analysis (Wang Y, Yan P, Fu T, Yuan J, Yang G, Liu Y, Zhang ZJ. The association between gestational diabetes mellitus and cancer in women: A systematic review and meta-analysis of observational studies. Diabetes Metab. 2020 Nov;46(6):461-471. doi: 10.1016/j.diabet.2020.02.003). Given that living conditions and nutrition can change significantly and that developing diabetes mellitus can also increase the risk of developing pancreatic cancer, it is necessary to review the risk of developing pancreatic cancer after gestational diabetes mellitus.

Two thirds of the reviewed literature, which covers various aspects of gestational diabetes mellitus and its consequences, was published no more than five years ago. There was no evidence of self-citation.

The examination of articles reveals data on the incidence of pancreatic cancer in women who had gestational diabetes mellitus during pregnancy, which is consistent with earlier publications on the subject.

Figure 1, in my opinion, duplicates the context-specific facts without adding any new information.

The following are some drawbacks of the work:
1) Line 51: glucose tolerance test (OGTT) should be changed to 22.3 per 100,000; 2) Line 119: oral glucose tolerance test (OGTT) should be changed to OGTT; 3) Table captions (1-6) are typically positioned above the table and remarks are provided at the bottom of the tables; 4) There are punctuation mistakes (there is no space between sentences in line 141, after the word in line 220, and between the link and the word that comes before it in lines 214, 221, 231, 236, 287, 303, 308);

5) When writing their surnames and initials in reference to their work, authors should follow a standard format: either place a period after the initials or leave it off (line 214, 221, 231, 236, 244, 249, 250, 267, 269, 274, 280, 284, 287, 293, 296, 298, 300, 302, 303, 305, 308).

 A detailed explanation of the process for topic-based article searching, sorting, and techniques for cross-referencing data should be included in the work's benefits.

Author Response

Dear Reviewer,

Thank you for your comments. 

The corrections have been made and highlighted in the main-text.

  1. Figure 1 was added to summarise and visualise the “1.1.1 Risk Factors of Pancreatic Cancer” stated in the main-text rather than adding additional information.
  2. Glucose tolerance test (OGTT) has been changed to changed to 22.3 per 100,000.
  3. Oral glucose tolerance test (OGTT) has been changed to OGTT.
  4. Table captions (1-6) have been positioned above the tables.
  5. Punctuations mistakes noted were corrected.
  6. Surnames and Initials in Reference have been corrected.

Once again thank you for your time.

Reviewer 2 Report

Comments and Suggestions for Authors

Overall a useful publication about the risk of GDM and pancreas cancer.  What to do with this could be added to the discussion, or discussion about what screening should or could be offered given this risk.

In the abstract the conclusion or results should include a Hazard Ratio and ideally a life time risk, is it still 7.1 life time risk?  That is high, should screening strategies be implemented and/or genetic testing to help determine who should be screened?  (see NCCN publication regarding the PRECEDE study).

Introduction:

Please state the risk in % of developing pancreatic cancer in women overall as compared to 7.1%.

Risk Factors

Line 61, I would not consider chronic pancreatitis a modifiable risk factor.

Line 62, where I would consider gut microbiome as a possibly modifiable risk factor.

Line 63, is cirrhosis a risk factor for pancreas cancer directly, or is it related to alcohol usage?  Is cirrhosis a modifiable or unmodifiable risk factor?

Line 77-78, it should be “Maturity Onset diabetes of the young” MODY.

Lines 78-79, syndromes normally are capitalized, I think.

Line 80 sets up a list “The 3 most common types” but then lists one, describes it and ends the sentence.  This is confusing structure.

Line 97, “resulting in slower gastric….”

Line 188, change to “a 100 gm” (gr is not standard SI for gram.”

Line 122, the sentence ends here after starting above with “Firstly” is not a complete sentence.

Line 127, “can result _in_ preterm births”

Section 1.1.4 when discussing risks of GDM to the infant, unless it causes pancreas cancer later in life, is not relevant to this article. 

Section 2.2, why only open access publications?  Why no original research if it exists?  Given the Open Access restriction, I believe Figure 4 should be clear on the number of articles excluded for lack of bring open access.

Section 4.1 lines 273-276 are confusing as written.  I think it is saying except from obesity and increase maternal age, PCOS, FH of T2DM, pre DM and previous fetal death increase the risk of GDM?

Section 4.2 focuses a lot on non GDM risk.

Comments on the Quality of English Language

Overall good, with some edits and review needed described above.

Author Response

Dear Reviewer,

Thank you for your comments. 

The corrections have been made to the main text with comments aside. 

  1. The "percent" has been change to "%".
  2. The "chronic pancreatitis" has been added to non-modifiable.
  3. The "gut microbiome" has been added to modifiable.
  4. Cirrhosis has been added to the non-modifiable section as its non-reversible (caused by alcohol consumption).

  5. Maturity Onset diabetes of the young” MODY has been corrected.
  6. "Syndromes" have been capitalised and highlighted throughout the text.
  7. The "3 most common types” has been rephrased and made more clear. 
  8. The “resulting in slower gastric" has been rephrased.
  9. 100 "gm" instead of "gr" has been corrected.
  10. The "Firstly" has been completed.
  11. The "can result in preterm birth" has been corrected.
  12. Section 1.1.4 has been added as supplementary data.

  13. The reason open access publications were used were due to the availability of the full article for examination and not only of the abstract.

  14. The sentence states that not only obesity but additionally to obesity, PCOS as well as family history of T2DM, prediabetes, previous history of fetal death can also play an important role in the development of GDM.Once again thank you for your feedback and your time.